# New Insights into Potocki-Shaffer Syndrome: Report of Two Novel Cases and Literature Review

**DOI:** 10.3390/brainsci10110788

**Published:** 2020-10-28

**Authors:** Slavica Trajkova, Eleonora Di Gregorio, Giovanni Battista Ferrero, Diana Carli, Lisa Pavinato, Geoffroy Delplancq, Paul Kuentz, Alfredo Brusco

**Affiliations:** 1Department of Medical Sciences, University of Torino, 10126 Turin, Italy; slavica.trajkova@unito.it (S.T.); lisa.pavinato@unito.it (L.P.); 2Medical Genetics Unit, Città della Salute e della Scienza, University Hospital, 10126 Turin, Italy; eleonora.digregorio@unito.it (E.D.); 3Department of Public Health and Paediatrics, University of Torino, 10126 Turin, Italy; giovannibattista.ferrero@unito.it (G.B.F.); diana.carli@unito.it (D.C.); 4Centre de Génétique Humaine, Université de Franche-Comté, 25000 Besançon, France; gdelplancq@chu-besancon.fr (G.D.); 5Service de Pédiatrie, CHU, 25000 Besançon, France; 6Oncobiologie Génétique Bioinformatique, PCBio, Centre Hospitalier Universitaire de Besançon, 25000 Besançon, France; pkuentz@chu-besancon.fr (P.K.); 7UMR-Inserm 1231 GAD, Génétique des Anomalies du développement, Université de Bourgogne Franche-Comté, 21000 Dijon, France; 8Fédération Hospitalo-Universitaire Médecine Translationnelle et Anomalies du Développement (FHU TRANSLAD), Centre Hospitalier Universitaire de Dijon et Université de Bourgogne Franche-Comté, 21000 Dijon, France

**Keywords:** Potocki-Shaffer, infantile spasms, *PHF21A*, SCNA, LSD-CoREST, epileptic encephalopathy, west syndrome, intellectual disability

## Abstract

Potocki-Shaffer syndrome (PSS) is a rare non-recurrent contiguous gene deletion syndrome involving chromosome 11p11.2. Current literature implies a minimal region with haploinsufficiency of three genes, *ALX4* (parietal foramina), *EXT2* (multiple exostoses), and *PHF21A* (craniofacial anomalies, and intellectual disability). The rest of the PSS phenotype is still not associated with a specific gene. We report a systematic review of the literature and included two novel cases. Because deletions are highly variable in size, we defined three groups of patients considering the PSS-genes involved. We found 23 full PSS cases (*ALX4*, *EXT2,* and *PHF21A*), 14 cases with *EXT2-ALX4*, and three with *PHF21A* only. Among the latter, we describe a novel male child showing developmental delay, café-au-lait spots, liner postnatal overgrowth and West-like epileptic encephalopathy. We suggest PSS cases may have epileptic spasms early in life, and *PHF21A* is likely to be the causative gene. Given their subtle presentation these may be overlooked and if left untreated could lead to a severe type or deterioration in the developmental plateau. If our hypothesis is correct, a timely therapy may ameliorate PSS phenotype and improve patients’ outcomes. Our analysis also shows *PHF21A* is a candidate for the overgrowth phenotype.

## 1. Introduction

Structural genetic variation is a class of sequence alterations typically spanning more than 1 kb [1]. They include quantitative variations such as copy number variations (CNVs), the most prevalent type of structural variation, and other alterations, including chromosomal rearrangements. They can span from thousands to millions of bases whose copy number varies between different individuals, and are the result of DNA gains or losses [2,3].

CNVs can be responsible for genomic disorders, Mendelian diseases associated with large gains, and losses of genetic material [4]. A number of well-delineated genomic disorders are presently known and can be divided into two main categories: recurrent genomic disorders, which span the same region and originate independently de novo in different patients, and the non-recurrent genomic disorders, whose extension is different in each patient although the disease-associated gene(s) is/are always included. Recurrent rearrangements are caused by nonallelic homologous recombination (NAHR) between flanking low-copy repeats (LCRs), or repetitive elements such as LINE and HERV elements [5,6]. Non-recurrent rearrangements are caused by more complex mechanisms, such as non-homologous end joining (NHEJ), fork stalling, and template switching (FoSTeS)/microhomology-mediated BIR (MMBIR) [7]. Their breakpoints are therefore variable hampering the identification of the causative gene associated with the disease.

Genomic disorders can include both (i) contiguous gene deletions or duplications as seen in Williams-Beuren syndrome (MIM# 194050), where more than one causative gene is present [8] or (ii) CNVs of genes or portions of genes (exons) leading to Mendelian disorders (e.g., Rubinstein-Taybi syndrome; MIM# 180849) [9].

The main molecular mechanism is related to changes in dosage sensitive genes, in other words, genes whose dosage is critical to determine a healthy phenotype. Both deletions leading to haploinsufficiency or duplications leading to triplosensitivity can lead to a phenotype [10]. The identification of causative genes within the critical region of a genomic disorder is complex. It can rely on the availability of haploinsufficiency scores provided by the GnomAD consortium [11], and the detection of several affected subjects with a deleted/duplicated region showing overlap on a few candidate genes.

We focused our attention on the Potocki-Shaffer syndrome (PSS), a rare non-recurrent contiguous gene deletion syndrome mapping on 11p11.2 (MIM# 601224) [12,13]. The classical phenotype comprises multiple exostoses, biparietal foramina, and neurodevelopmental delay as cardinal features. Craniofacial abnormalities, epilepsy, tapering fingers, eye and hearing abnormalities, hypothyroidism, immunodeficiency, and genital malformations in males have also been reported [14].

Current literature implies a critical minimal region with haploinsufficiency of three genes: aristaless-like homeobox 4 (*ALX4*, MIM* 605420), exostosin 2 (*EXT2*, MIM* 60821), and PHD finger protein 21A (*PHF21A*, MIM* 608325). In rare cases, the deletion extends centromerically causing overlap with the WAGR syndrome (Wilms’ tumor, aniridia, genitourinary anomalies/gonadoblastoma and mental retardation) [15,16,17]. Haploinsufficiency for *PAX6* (causing aniridia, MIM# 106210) [18] and *WT1* genes (predisposing Wilms’ tumor, genital abnormalities, and nephropathies, MIM# 194072) [19] lead to the main WAGR features.

Here, we report a critical revision of the literature grouping describing patients in three categories based on the presence of the PSS-associated genes within the deletion (*ALX4*, *EXT2*, and *PHF21A*; *EXT2-ALX4*; and *PHF21A* only). We also describe a novel PSS female patient (Decipher 286390) and the third smallest *de novo* 11p11.2 microdeletion spanning *PHF21A* (Decipher 415213) in a male with developmental delay (DD), intellectual disability (ID), café-au-lait spots, liner postnatal overgrowth, and West-like pharmacoresistant epilepsy. These latter cases are instrumental to better define the role of *PHF21A* in the phenotype.

## 2. Materials and Methods

Genomic DNA was isolated from peripheral blood using a standard procedure (Qiagen, Hilden, Germany) and quantified by Nanodrop spectrophotometer (Thermo Scientific, Waltham, MA, USA). We performed array-CGH with a 60 K whole-genome oligonucleotide microarray following the manufacturer’s protocol. Slides were scanned using a G2565BA scanner (Agilent Technologies, Santa Clara, CA, USA) and analyzed using CytoGenomics version 5.0.2.5 (Agilent Technologies, Santa Clara, CA, USA) with the statistical algorithm ADM-2 and a sensitivity threshold of 6.0. At least three consecutive aberrant probes identified significant copy-number changes. We compared our findings to known CNVs listed in the Database of Genomic Variants (DGV, http://projects.tcag.ca/variation) and in the DECIPHER database (https://decipher.sanger.ac.uk/). TaqMan real-time quantitative PCR (qPCR) analysis (was used to measure copy number variants at 11p11.2 in genomic DNA on the gene *PEX16* (NM_057174.3, exon 11), primers 5′-cagagcctggtgaacagtga; 5′-aggatgcagggcttaaagtg; #36 UPL probe (Roche Diagnostics, Risch-Rotkreuz, Switzerland); with RnaseP reference gene, VIC-labeled pre-designed TaqMan gene expression assays (P/N 4316844, Applied Biosystems, Thermo Fisher Scientific, Waltham, MA, USA). We carried out the reaction with an ABI 7500 Fast real-time PCR machine using the ABI TaqMan Universal PCR master mix according to the manufacturer’s instructions (Applied Biosystems, Thermo Scientific, Waltham, MA, USA). Efficiencies of the assays were similar and in a range of 90–110%. Samples from affected individuals and unrelated healthy controls were run in triplicate. The mean Ct value was used for calculations using the ΔΔCt method [20].

All subjects gave their informed consent for inclusion before participation. The study was conducted in accordance with the Declaration of Helsinki.

We identified all potentially relevant articles, limited to English-language studies by searching PubMed (https://pubmed.ncbi.nlm.nih.gov/), and an additional hand search of the reference lists from the obtained articles. We considered abstracts at international conferences only if they reported relevant cases. If some crucial information were not provided or they were not feasible at the time of writing the article, all eligible authors were contacted, with a second mail as a reminder if responses were not received. The search was performed up to December 2019 with an up-to-date e-alert from each search platform.

We extracted the characteristic and additional phenotypic elements from each study. Furthermore, we concentrated on describing the dysmorphology traits, reporting the most frequent features. If the growth parameters [21] were accessed in several time periods after birth, only the last one was included. An array-CGH and newer data were preferred over FISH examinations.

In order to construct one scheme of all reported PSS deletions, we have converted the FISH-BAC clones in the GRCh37/hg19 using their distal coordinates as a minimal zone of the deletion (further details in Appendix A).

## 3. Results

### 3.1. Case Report 1 (Decipher 286390)

She is the unique child born by caesarian section from non-consanguineous European parents (father from France and mother of Spanish origin). Her prenatal course was uncomplicated. The family history was negative for other individuals with neurodevelopmental disorders except for a cousin of the mother who had a son with mild to moderate intellectual disability. At birth, neonatal hypotonia was noticed and a cerebral CT-scan revealed a thin corpus callosum. Multiple exostoses were reported. Epilepsy started at the age of 8 years, and it was initially treated with lamotrigine. At the age of 12 years, she presented a status epilepticus secondary to *Haemophilus influenzae* septic shock associated with acute respiratory distress syndrome (ARDS) (detailed description in Appendix A). On the last visit (patient is now in her twenties) the epilepsy was under control with a cocktail of anti-epileptic drugs.

She had a severe global developmental delay. She crawled on all four limbs but did not master chair-to-floor transfers and the use of the wheelchair. She could walk with difficulty using a walker. She has a quick and efficient prehension, but she does not have the resources for instrumental use of objects. In terms of language, she can say several single words but not structured sentences. She has sleep apnea and an IgA immune deficiency treated by injection of gamma globulins, discovered at 12 years old during her hospitalization. No known gene in the deleted interval is associated with immunodeficiency.

At birth, her growth parameters were weight 2970 g (−0.87 SD), height 51 cm (+0.56 SD), and occipoto frontal circumference (OFC) 34 cm (−0.54 SD). At 12 years, weight 31 kg (−3 SD), height 134 cm (−2 SD), body mass index (BMI) 17 kg/m^2^ (25–50th centile). At 20 years, weight 37 kg (−3 SD), height 141 cm (−3.4 SD); BMI 18.6, 17 kg/m^2^ (10–25th centile), and OFC 53 cm (−1.20 SD). Her dysmorphic features are described on Figure 1.

The array-CGH detected a deletion overlapping the PSS critical region with a min-max size of 8–9.1 Mb (array (GRCh37) 11p11.12 (42272129x2, 42333416_50378802x1, 51379160x2)). The inheritance of the deletion could not be tested. FISH analysis confirmed a deletion 11p11.2p11.2 (RP11-70A9-RP11-465I24), in line with previous array-CGH data.

### 3.2. Case Report 2 (Decipher 415213)

Patient was a male born at term (Apgar 9/9, 42nd gestational week with vaginal delivery; birth weight 4070 g (>85th centile); length 52 cm (50th centile); head circumference 33 cm (<15th centile)) from non-consanguineous healthy parents. He was hospitalized at 10 months of age due to apyretic convulsive crises followed by hypotonia, somnolence, and developmental delay. His mother reported daily episodes of short duration with quick muscle contraction of the arms, and a fixed look as if the “child was scared”. There was no family history of malformation, epilepsy, or developmental delay.

The electroencephalography (EEG) (awake/sleep) in the first days of his hospital admission demonstrated an interictal hypsarrhythmia characterized by slow chaotic high voltage delta and theta activity (prominent in the right temporo-parietal region), mono/polymorphic preceded by multifocal frequent bouffés type PO, PPO spike complexes (Figure 2). Due to an EEG pattern indicative of West syndrome, the patient was administrated ACTH, cortisone, and phenobarbitone. After the arterial hypertension, most likely a side effect from the ACTH, a diuretic and calcium-antagonist were prescribed.

Beside the Magnetic Resonance Imaging (MRI) signs of benign external hydrocephalus and bilateral mastoiditis with modest micellar sinusopathy and minimal pericardial effusion on electrocardiogram. All other examinations were in normal ranges (sensory evoked potentials, metabolic profiles, echo and computer tomography (CT) scan).

Currently (5 years of age), the patient presents a linear postnatal overgrowth with severe intellectual disability (weight 23 kg (90–97th centile), height 121 cm (>97th centile; father 176 cm, mother 170 cm), and head circumference of 53.5 cm (90–97th centile)).

Using array-CGH, we detected a *de novo* 323–472 kb microdeletion, partially overlapping the PSS critical region (arr(GRCh37) 11p11.2(45553929x2,45670806_45993729x1,46027199x2)dn) The deletion was confirmed by real-time quantitative PCR (qPCR), and encompassed eight protein coding genes: *C11orf94*, *CHST1*, *CRY2*, *GYLTL1B*, *MAPK8IP1, PEX16*, *PHF21A*, and *SLC35C1*.

### 3.3. Review of Reported PSS Cases

We selected 18 published articles for a total of forty 11p11.2 deletions, including the two reported here [12,13,15,16,17,22,23,24,25,26,27,28,29,30,31,32,33,34]. All are represented in Figure 3 using the UCSC Genome Browser custom track (Appendix A). In three, the WAGR critical region was included [15,16,17]. We excluded all PSS cases carrying an additional pathogenic deletion or duplication [35,36]. Reports of the patients with heterozygous deletions which appeared as early as 1977 [37] or those where no sufficient data was provided even after contacting the authors [38] were also excluded.

Detailed clinical data including dysmorphology was obtained for each patient when available (Appendix A).

We noted that the deletions described are highly variable. To perform a better genotype-phenotype correlation, we divided cases into three groups based on the involvement of the three PSS critical genes. We found 23 full PSS cases (*ALX4*, *EXT2*, and *PHF21A*), 14 cases with *EXT2-ALX4*, and three with only *PHF21A*. The minimal deletion reported by Chuang et al. [24] did not affect *ALX4/EXT2*; however, the patients had biparietal foramina and multiple exostosis, and thus were classified as complete PSS deletion.

Half of the patients (20/40) had reports of their birth parameters (Table 1). The majority had weight/height or head circumference adequate for their age (15/20); three were small, and two were large for their gestational age at birth (>90th centile or 2 SD above the mean weight, length, or both). Postnatal overgrowth was observed in four patients, and two did not catch up to expected parameters despite the normal parameters at birth.

Developmental delay was noted in half of the patients (22/40); isolated language delay in one, additional autistic traits in patient 3 by Wuyts et al. [25], and in the follow-up of patient III-1 by Shaffer et al. [13] (Table 1; Appendix A).

At the neurologic examination, hypotonia was the most prevalent finding detected in 16 patients, although the definition was not consistent. We found epilepsy in 14 cases, clinically very heterogeneous by etiology and definition (Table 1). Notably, all except one showed severe deterioration on the developmental plateau and intellectual disability (Table 1).

Various brain anomalies were seen on MRIs such as thin (n = 5), absent (n = 2), or hypoplastic corpus callosum (n = 1) (Table 1; Appendix A). Prominent cerebrospinal fluid spaces were detected in six patients (Table 1). Other brain anomalies were dysplasia/hypoplasia of the cerebellar cortex/vermis (five patients), choroid plexus cyst (two patients), and an unusual report of meningoencephalocele (Table 1; Appendix A).

Genitourinary anomalies such as micropenis (9/28) and cryptorchidism (8/28) were frequent findings in males. Strabismus (12/40) and nystagmus (5/40) were prevailing from the ocular anomalies (Table 1; Appendix A).

Among 28 patients with dysmorphic features, brachycephaly was commonly reported in 17 patients (61%), followed by broad forehead and epicanthus present in 12 (43%), followed by downturned mouth, prominent nasal bridge, high forehead, sparse lateral eyebrows, and short philtrum (Table 2). Tapering fingers and brachydactyly were hand anomalies reported in a minority (Appendix A).

## 4. Discussion

In contiguous genes syndromes, the identification of the causative genes responsible for the phenotype relies on the availability of patients with different but overlapping deletions/duplications and on functional or genetic data of the genes spanning the CNV. In Potocki-Shaffer syndrome, three genes are reported to be causative: *ALX4*, *ELX2*, and *PHF21A*. We aimed at improving genotype-phenotype correlation on the syndrome and revised literature data collecting a total of 40 cases.

Patients with a deletion involving *ALX4*, *ELX2*, and *PHF21A* genes had the cardinal PSS features: biparietal foramina, multiple exostosis, and intellectual disability and craniofacial anomalies associated with *ALX4*, *ELX2*, and *PHF21A*, respectively. The haploinsufficiency of *ALX4* [39] was also suggested to explain the micropenis in some of the males with PSS.

*SLC35C1* [23], *PEX16* [23], and *GYLT1B* [40,41] have been proposed as candidate genes for the hypotonia reported in most of the PSS patients. Actually, these genes are not haploinsufficient (pLI scores close to 0 [42]), and they unlikely can contribute to this condition.

We were particularly interested in the role of *PHF21A* in PSS. Disrupting variants hitting *PHF21A*, such as balanced translocations [22], truncating [43], or missense variants [44] are reported in patients with intellectual developmental disorder and craniofacial dysmorphism. Notably, *PHF21A* disruption has also been associated with hypotonia, and different types of epilepsy among which West-like epileptic encephalopathy [43]. West syndrome (WS) [45] includes a triad composed of infantile spasms (epilepsy) [46], hypsarrhythmia (EEG pattern), and intellectual disability. The infantile spasms of WS are subtle and short, they disappear by the age of 2 years and are often overlooked if not recognized by trained and experienced eyes. Their early diagnosis and a shorter delay to treatment are essential for evading their long term morbidity such as intellectual disability or a more severe type of epilepsy [47].

We noted that West syndrome was also diagnosed in patient 1 (Decipher code 415213), at the initial stage of disease (10 months of age), suggesting this feature is present in early phases of PSS. We analyzed the reports of epilepsy/seizures in the whole PSS literature cohort, but we were unable to find further cases with infantile spasms/WS. We think that the age of the probands at diagnosis is likely the most relevant explanation, because of the early onset of infantile spasms and subtle clinical presentation [13,34]. In some cases, a poor clinical description [23], or different deletion sizes [29] may have hidden the WS features. On the other hand, we found two neonates, while searching infantile spasms among cases with *PHF21A* point mutations (one truncating and one missense variant) [43,44]. Our observation needs future clinical confirmation; however, if infantile spasms are truly present in the early phases of PSS and remain untreated, they could lead to intellectual disability or to more severe types of epilepsy described in this disease. Since *PHF21A* strongly correlates with intellectual disability, we suggest that it could also be the leading gene for the infantile spasms and epilepsy.

The histone methyl reader protein (BHC80) [48] encoded by *PHF21A* is best known for its role in regulation of a huge number of neuronal genes during embryogenesis [49,50], and it is particularly important in the development of nerve cells [51] and bone structures of the face [22]. It is a well conserved gene and is highly intolerant to variation (pLI = 1.0; Z-score = 2.86, GnomAD ver.2.1.1 http://gnomad.broadinstitute.org/). This reader protein is part of LSD-coREST [52] complex and recognizes the epigenetic marks on core H3K4 through specialized motifs [53], which researchers speculate helps keep the histone demethylated and the genes turned off (repressed) [53,54]. These target genes have a specific cis-regulatory elements known as repressor element-1 (RE1) or neural restrictive silencer (NRS) [55]. Among them are SCN2A [52] or SCN3A [22,56], sodium channel encoding genes, strongly associated with epileptic encephalopathy [57]. The *PHF21A* haploinsufficiency is indeed altering the expression of these genes, as it was demonstrated in patients’ derived *PHF21A* haploinsufficient cell lines [22,56].

Concerning dysmorphological traits such as brachy/microcephaly and mild micrognathia, the role for *PHF21A* was supported by the generation and rescue of a zebrafish model, where *PHF21A* orthologous suppression produced abnormalities in the development of the head, face, and jaw [22]. Yet, these features were not present in all patients with a deletion affecting *PHF21A*, suggesting incomplete penetrance.

Birth/postnatal overgrowth, present in some of the PSS cases, could also be associated with *PHF21A* haploinsufficiency. A recent report on disorders associated to overgrowth with intellectual disability (OGID) showed that 14 genes were involved [58]. Perturbation of epigenetic regulation was the main pathogenic mechanism. Interestingly, most of these genes have motifs recognized by the LSD-coREST complex where *PHF21A* is taking part [59,60].

## 5. Conclusions

We are presenting a detailed systematic review of all reported PSS cases, including two novel ones. The phenotype is confirmed heterogeneous, but we highlighted the possibility that infantile spasms are present before 2 years of age. We highlight the importance of confirming this observation by prompt examination of PSS cases. An early diagnosis of infantile spasms may shorten the delay to treatment and subsequently lower or even abolish the risk of intellectual disability [61,62]. This goal could be achieved in case of a suspect or in families at risk by:-a detailed clinical examination of neonates, particularly focused on getting a full neurological assessment-a complete video-EEG recording-a magnetic resonance (MR) study of the brain

We also suggest the inclusion of *PHF21A* into gene panels for infantile spasms, performing prompt genetic testing and EEG in suspected patients, and in probands from at-risk families.

Web Resources

Decipher: http://decipher.sanger.ac.ukWorkingsgnomAD, http://gnomad.broadinstitute.org/OMIM, http://www.omim.org/UCSC Genome Browser, https://genome.ucsc.edu/

## Figures and Tables

**Figure 1 brainsci-10-00788-f001:**
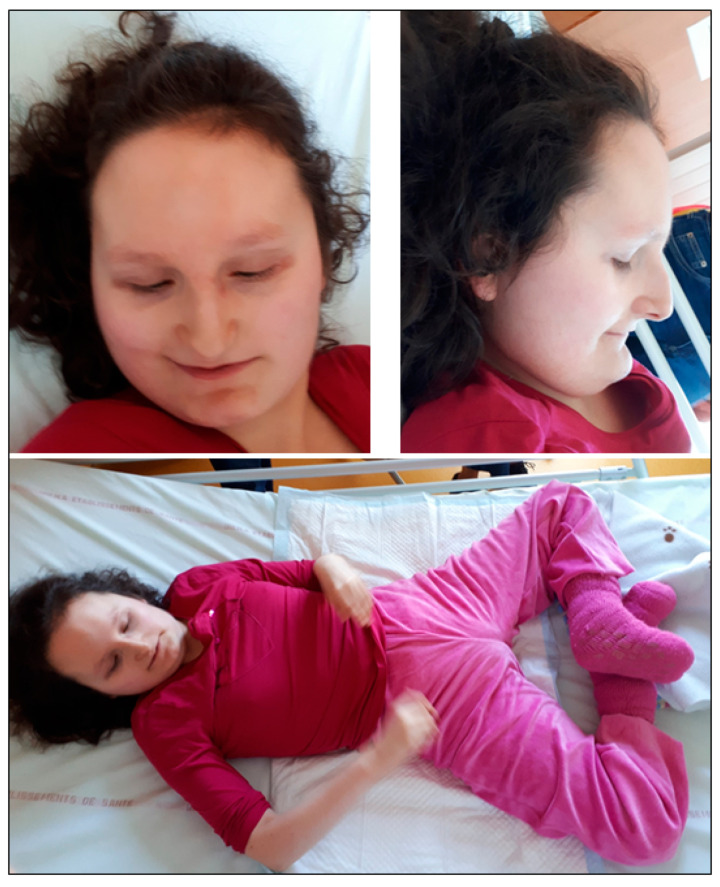
Photography of patient 1 (Decipher 286390) at 20 years old. She has a high and broad forehead, sparse lateral eyebrows, long nose with prominent nasal bridge, short and smooth philtrum, thin lips, prominent chin with horizontal crease, quite large mouth, and a short neck. There is a large abduction and an external rotation of the two hips.

**Figure 2 brainsci-10-00788-f002:**
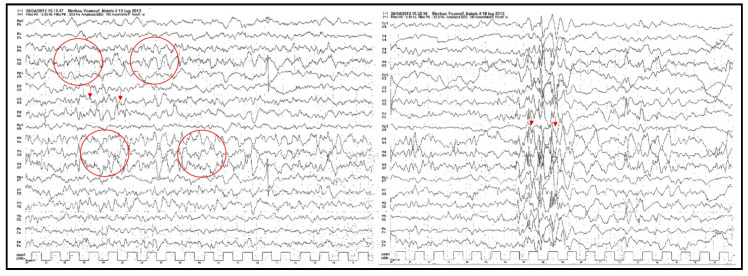
Electroencephalography (EEG): interictal hypsarrhythmia. Slow high voltage of delta and theta activity (circles on the left), mono/polymorphic preceded by multifocal PO (arrows on the left), PPO spike (arrows on the right).

**Figure 3 brainsci-10-00788-f003:**
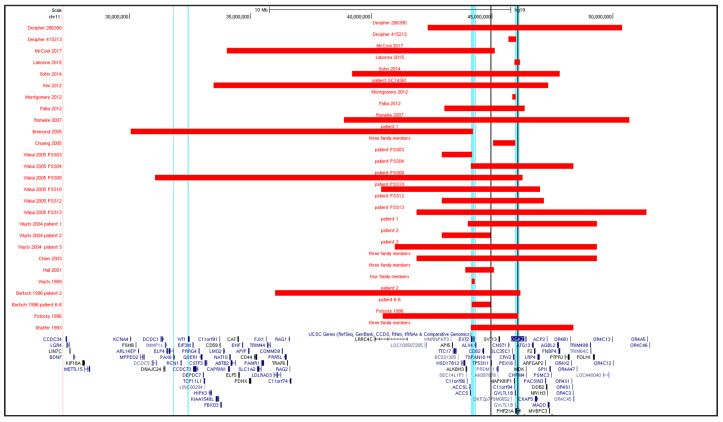
Scheme for the 40 known deletions encompassing the Potocki-Shaffer syndrome (PSS) critical region. In red, the extension of the minimal deleted region. Vertical light blue lines indicate the borders of the critical PSS region, spanning the *PHF21A*, *ALX4*, and *EXT2* genes. Black vertical lines indicate the redefined critical region for ID/DD in PSS: a ~1.1 Mb region containing 12 annotated genes between markers D11S554 and D11S1319 [22]. On the left side, the *PAX6* and *WT1* genes associated with Wilms’ tumor, aniridia, genitourinary anomalies/gonadoblastoma and mental retardation syndrome (WAGR). Note that some of the red bars indicate more than one case in the same family.

**Table 1 brainsci-10-00788-t001:** Clinical characteristic of PSS cases reported in the literature.

	Decipher 286390	Decipher 415213		McCool	Labonne		Sohn		Kim GC14361		Montgomery	Palka	Romeike	Bremond	Chuang Patient 1	Chuang Patient 2	Chuang Patient 3		Wakui PSS03	Wakui PSS04	Wakui PSS08	
**PSS**	**+**						**+**		**+**			**+**	**+**		**+ ^&^**	**+ ^&^**	**+ ^&^**			**+**	**+**	
**ALX/EXT4**				**+**										**+**					**+**			
**PHF21A**		**+**			**+**						**+**											
Birth parameters																						
SGA/Undergrowth	-	-		-	-		+		n.a		-	-	n.a	n.a	n.a	n.a	n.a		n.a	n.a	n.a	
Appropriate	+	+		+	-		-		n.a		-	+	n.a	n.a	n.a	n.a	n.a		n.a	n.a	n.a	
LGA/Overgrowth	-	-		-	+		-		n.a		+	-	n.a	n.a	n.a	n.a	n.a		n.a	n.a	n.a	
Postnatal growth																						
Undergrowth	+	-		-	-		+		n.a		-	-	n.a	- †	n.a	n.a	n.a		+	+	n.a	
Appropriate	-	-		-	-		-		n.a		-	+	n.a	+	n.a	n.a	n.a		-	-	n.a	
Overgrowth	-	+		+	+		-		n.a		+	-	n.a	-	n.a	n.a	n.a		-	-	n.a	
Neurodevelopment																						
Developmental delay	+	+		+ »	+		+		+		+	+	+	-	+	-	-		-	+	+	
Intellectual disability	+	+		-	-		+		-		+	+	+	+	-	+	+		-	-	-	
Language delay	-	-		-	-		-		-		-	-	-	-	-	-	-		-	-	-	
Neurological findings																						
Hypotonia	+	-		-	-		+		+		+	-	-	-	-	-	-		-	+	-	
Epilepsy	+	+		-	-		-		- °		-	+	+ *	-	-	+	+		-	-	-	
MRI findings																						
Corpus callosum	+	-		-	-		-		-		-	-	+	-	-	-	-		-	-	+	
Prominent CSF spaces	-	-		-	-		-		-		-	-	+	-	-	-	-		-	-	-	
Other brain anomalies	-	+		-	-		-		-		-	-	+	-	-	-	-		-	-	-	
Genitourinary																						
Micropenis	-	-		-	+		-		-		-	-	-	-	+	-	-		-	+	-	
Cryptorchidism	-	-		-	-		-		+		-	-	+	-	-	-	+		-	+	-	
Ocular anomalies																						
Cataract	-	-		-	-		-		-		-	-	-	+	-	-	-		-	-	-	
Strabismus	-	-		+	-		+		-		-	-	-	-	+	-	-		-	-	-	
Nystagmus	-	-		-	-		+		-		-	-	-	+	-	-	-		-	-	-	
Hearing anomalies																						
hearing loss	-	-		-	-		-		+		-	-	-	-	-	-	-		-	-	-	
Other	IgA deficiency; sleep apnea	Café-au-lait spots		Recurrent infections			Ptosis		Pectus excavatum. Recurrent otitis media		Pectus excavatum.		Cardiomyopathy; Osteochondromas; Anemia	Bilateral aniridia; Kidney tumor; Obesity		Umbilical hernia			Bowing of lower extremities	Umbilical hernia; Myopia; Recurrent infections	Wilms’ tumor; Aniridia	
	Wakui PSS10	Wakui PSS12	Wakui PSS13		Wuyts patient 1	Wuyts patient 2		Wuyts patient 3			Chien 3 patients	Hall 3 patients	Wuyts 4 patients	Bartsch Patient 2	Bartsch Patient 6–8	Potocky	McGaughran	Shaffer III-1		Shaffer III-2	Shaffer II-4	
**PSS**	**+**	**+**	**+**		**+**					**+**	**+**			**+**		**+**	**+**	**+**		**+**		**+**
**ALX/EXT4**						**+**						**+**	**+**		**+**							
**PHF21A**																						
Birth parameters																						
SGA/Undergrowth	n.a	-	+		-	-				-	n.a	n.a	n.a	-	-	-	-	+ ^		n.a		-
Appropriate	n.a	+	-		+	+				+	n.a	n.a	n.a	+	+	+	+	-		n.a		+ ^
LGA/Overgrowth	n.a	-	-		-	-				-	n.a	n.a	n.a	-	-	-	-	-		n.a		-
Postnatal growth																						
Undergrowth	-	-	+		-	-				+	n.a	n.a	n.a	-	-	-	-	+ ^		n.a		-
Appropriate	+	+	-		+	+				-	n.a	n.a	n.a	+	+	+	+	-		n.a		+ ^
Overgrowth	-	-	-		-	-				-	n.a	n.a	n.a	-	-	-	-	-		n.a		-
Neurodevelopment																						
Developmental delay	-	+	+		+	+				+	+	-	-	-	-	+	-	+ ^		-		-
Intellectual disability	-	-	-		+	+				+	+	-	-	+	-	+	+	+		+		+
Language delay	+	-	-		+	-				-	-	-	-	-	-	-	-	+ ^		-		-
Neurological findings																						
Hypotonia	-	-	+		+	+				+	+	-	-	+	-	-	-	+		+		+
Epilepsy	-	-	-		+	-				+	- º	-	+ ΅	+	-	-	-	+ ^		-		+
MRI findings																						
Corpus callosum	+	+	-		-	-				-	+	-	-	-	-	-	-	-		-		-
Prominent CSF spaces	-	-	+		-	-				-	+	-	-	-	-	-	-	+ ^		-		-
Other brain anomalies	-	+	-		+	-				+	-	-	+ ΅	-	-	-	-	+ ^		-		+
Genitourinary																						
Micropenis	+	-	-		-	-				+	+ º	-	-	+	-	-	+	-		-		+
Cryptorchidism	-	-	+		-	-				-	- º	-	-	-	-	-	+	-		-		+
Ocular anomalies																						
Cataract	-	-	-		-	-				-	-	-	-	-	-	-	+	-		-		-
Strabismus	+	-	+		-	+				+	+	-	-	+	-	+	-	+		-		+
Nystagmus	-	-	-		-	+				+	-	-	-	+	-	-	-	-		-		-
Hearing anomalies																						
hearing loss	-	+	+		-	-				-	-	-	-	-	-		-	-		-		+ ^
Other	Deceased from multiorgan failure	Small testis	VSD; Recurrent infections		High myopia; Obesity	Anal atresia and fistula; VSD				Myopia; Acrocephalosyndactyly; Adipose		7625 Asthma; Hyperactivity		Adipose appearance	Café-au-lait spots Capillary hemangioma	Bilateral ptosis	Aniridia; Wilms’ tumor; Short stature	Borderline hypothyroidism; Simian crease		borderline high TSH; Simian crease; Obese		Adipose appearance; Aggressive behavior

Notes: SGA, small for gestational age; LGA, large for gestational age; n.a, not available; VSD, ventricular septal defect; », history of developmental delay; °, history of static encephalopathy; *, information for antiepileptic drugs; †, height 2.4 SD; &, minimal deletion does not affect the ALX/EXT4; ^º^, micropenis only in the index case; report of mother’s two brothers of febrile seizures and epilepsy; the older had right-side cryptorchidism, whereas the younger umbilical hernia; ΅, epilepsy only in the proband; CT performed only in the proband and his father (IV-1 and III-1); ^, additional information from Swarr et al., 2009 [14]. The first three rows indicate which of the three critical PSS genes is involved in the deletion: PSS indicates the deletion spans at least *ALX4*, *EXT2*, *PHF21A*.

**Table 2 brainsci-10-00788-t002:** Dysmorphology of PSS reported patients in the literature.

Dysmorphology	*n*	*%*	PSS	*ALX/EXT4*	*PHF21A*
Brachycephaly	17	61	14	2	1
Broad forehead	12	43	9	1	1
Epicanthus	12	43	9	1	2
Downturned mouth	11	39	11	0	0
High forehead	10	36	8	2	0
Prominent nasal bridge	10	36	10	0	0
Sparse lateral eyebrows	9	32	8	1	0
Short philtrum	9	32	7	1	1
Microcephaly	8	29	8	0	0
Hypoplastic nares	6	21	5	1	0
Broad nasal tip	5	18	5	0	0
Low set ears	5	18	4	1	0
Large/protuberant ears	5	18	3	1	1
Telecanthus	4	14	4	0	0
Upslanting palpebral fissures	4	14	4	0	0
Thin lips	4	14	3	0	1
Micrognathia	4	14	2	1	1
Turricephaly	3	11	2	0	1
Broad nasal bridge	3	11	2	0	1
Short neck	3	11	3	0	0
Small nose	2	7	2	0	0
Small mouth	2	7	1	0	1
Full cheeks	2	7	0	1	1
Prominent chin	2	7	1	1	0
Downslanted palpebral fissures	1	3	0	1	0
TOTAL	28				

Notes: PSS is Potocki-Shaffer syndrome.

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
