# Peer review of "New Insights into Potocki-Shaffer Syndrome: Report of Two Novel Cases and Literature Review"

_brainsci, 2020, doi:10.3390/brainsci10110788_

Round 1

Reviewer 1 Report

This manuscript reports two additional cases of patients with deletion 11p11.2, one of have having hypsarrhytmia, and make a review of the litterature. (The abormal movements reported by the mother are not decribed as spasms in the case report).

To my point of view there are not enought evidence to support the title, i.e. a major link between PHF21A and infantile spasms. As stated by the authors page 11, infantile spasms have been reported oocasionally among different other types of epilepsy. Furthermore, as shown in their table that provide a comprehensive summary of the litterature, epilepsy is far from constant in PHF21A mutations.

In addition, the paper is globally written as a review of the litterature about deletion 11p11.2, with paragraphs about cytogenetic mechanisms and other clinical features that are far from the title.

I think the authors have to chose between a review or a paper focused o a specific point, such as infantile spams. In this latter case, my opinion is that convincing evidence is not sufficient to support the conclusion.

Author Response

Reviewer 1

This manuscript reports two additional cases of patients with deletion 11p11.2, one of have having hypsarrhytmia, and make a review of the literature. (The abnormal movements reported by the mother are not described as spasms in the case report).

Answer: We tried to better translate mother’s words into the following sentence:” His mother reported daily episodes of short duration with quick muscle contraction of the arms, and a fixed look as the “child was scared”. “

To my point of view there are not enough evidence to support the title, i.e. a major link between PHF21A and infantile spasms. As stated by the authors page 11, infantile spasms have been reported occasionally among different other types of epilepsy. Furthermore, as shown in their table that provide a comprehensive summary of the literature, epilepsy is far from constant in PHF21A mutations.

In addition, the paper is globally written as a review of the literature about deletion 11p11.2, with paragraphs about cytogenetic mechanisms and other clinical features that are far from the title.

I think the authors have to choose between a review or a paper focused on a specific point, such as infantile spams. In this latter case, my opinion is that convincing evidence is not sufficient to support the conclusion.

Answer: We thank reviewer 1 for suggestions and comments. Concerning the title, with his/her comments and changed it into “New insights into Potocki-Shaffer Syndrome: report of two novel cases and literature review”. We also agree that data available are not strong enough to support our hypothesis for the role of PHF21A in infantile spasms. Therefore, we considered our article as a review of the literature with two further cases.

Reviewer 2 Report

The paper has reviewed 40 Potocki–Shaffer syndrome (PSS) cases that include all available reported cases and two novel cases. It has throughfully investigated all possible symptoms and causative gene deletions of PSS. In particular, it has focused on the roles of PHF21A in PSS. However, I have a concern with respect to the statistical significancy that ensures scientifically what it has found. Regarding 40 cases, it may be allowable to present frequency tables or summary statistics of genes involved in the deletion versus reported symptoms along with corresponding chi-square tests.

Author Response

Reviewer 2

The paper has reviewed 40 Potocki–Shaffer syndrome (PSS) cases that include all available reported cases and two novel cases. It has thoughtfully investigated all possible symptoms and causative gene deletions of PSS. In particular, it has focused on the roles of PHF21A in PSS. However, I have a concern with respect to the statistical significancy that ensures scientifically what it has found. Regarding 40 cases, it may be allowable to present frequency tables or summary statistics of genes involved in the deletion versus reported symptoms along with corresponding chi-square tests.

Answer: We thank reviewer 2 for his/her comments and suggestions. Following his/her reasoning, we agree that providing a summary statistic of the involved genes in the deletion versus reported symptoms is useful for the reader and it is providing more clear presentation of the results.

The most sensible approach to present this data was dividing the cases into three categories – deletion affecting the whole PSS minimal region, deletion affecting ALX4, EXT2 and deletion affecting PHF21A only.

We did not consider other genes in the PSS minimal region, because current data, such as pLi score or gene function are not supporting their involvement.

We have extended Table 2 including three additional columns, which report the dysmorphological features in the three above categories. A similar division was done for the clinical data presented in Table 1. Since this table is already at the journal limits in size, a supplementary Table 2 was created. We did not summarize the birth and postnatal parameters because of too many missing reports.

Following reviewer request, we calculated the chi-square, testing the presence of a phenotype with the reported genotype of the deletion in the three categories (Whole-PSS; ALX4, EXT2 and PHF21A). Because the figures were often below 5, we also calculated the probability derived from the Fisher’s exact test.

All analyses were done using STATA Version 13 software.

Even if we collected all available reported PSS cases, given the rarity of the syndrome, it was still difficult to obtain significant values. Below, we provide Table 2 (Dysmorphology) with statistical analyses. The latter were omitted in the main text. Indeed, observing the numbers derived from the chi2 and Fisher’s exact test, we realized that they are not consistent likely due to the small number of cases. The only exception was the report of “downturned mouth” and “prominent nasal bridge” that remained significant in both analyses. However, again number of reported cases are small. Overall, we think it is better to report data but not trying to force conclusions from statistical analysis using such small surveys.

Table 2.

Dysmorphology

PSS

EXT/ALX

PHF21A

Pearson chi2

Pr

Fisher

Brachycephaly

14

2

1

2.5654

0.277

 0.265

6

3

2

Epicanthus

9

1

2

 1.7986

0.407

 0.495

11

4

1

Broad forehead

9

2

1

 0.1653

 0.921

1.000

11

3

2

Downturned mouth

11

0

0

7.2471

 0.027

 0.027

9

5

3

Prominent nasal bridge

10

0

0

6.2222

  0.045

 0.044

10

5

3

High forehead

8

2

0

 1.8667

0.393

 0.679

12

3

3

Sparse lateral eyebrows

8

1

0

2.3251

 0.313

 0.551

12

4

3

Short philtrum

7

1

1

0.4148

0.813

 1.000

13

4

2

Microcephaly

8

0

0

 4.4800

0.106

 0.121

12

5

3

Hypoplastic nares

5

1

0

0.9758 

0.614

1.000

15

4

3

Broad nasal tip

5

0

0

2.4348

0.296

  0.580

15

5

3

Low set ears

4

1

0

 0.7304

0.694

 1.000

16

4

3

Large/protuberant ears

3

1

1

 0.6168

0.735

 0.754

17

4

2

Telecanthus

4

0

0

 1.8667

0.393

0.722

16

5

3

Upslanting palpebral fissures

4

0

0

1.8667

0.393

0.722

16

5

3

Thin lips

3

0

1

1.7306

 0.421

  0.485

17

5

2

Micrognathia

2

1

1

1.3222

0.516

 0.318

18

4

2

Turricephaly

2

0

1

 2.2151

0.330

 0.362

18

5

2

Broad nasal bridge

2

0

1

2.2151

0.330

 0.362

18

5

2

Short neck

3

0

0

1.3440

0.511

 1.000

17

5

3

Small nose

2

0

0

 0.8615

0.650

 1.000

18

5

3

Small mouth

1

0

1

3.6256

0.163

 0.233

19

5

2

Full cheeks

0

1

1

5.8872

 0.053

 0.074

20

4

2

Prominent chin

1

1

0

1.6154

0.446

 0.497

19

4

3

Downslanted palpebral fissures

0

1

0

4.7704

0.092

 0.286

20

4

3

total =28

20

5

3

Statical analysis was also performed for phenotypes in Table 1 (Clinal characteristics; see below). Again, our conclusion was that the survey of cases is not suitable for this analysis given the limited number of cases in each categories.

FULL

EXT/ALX

PHF21A

Pearson chi2

Pr

Fisher

Developmental delay

17

2

3

15.1553

0.001

0.000

6

12

0

Intellectual disability

17

2

2

 12.7371

0.002

 0.001

6

12

1

Language delay

3

0

0

2.3972

0.302

 0.430

20

14

3

Hypotonia

14

1

1

10.5271

0.005

 0.002

9

13

2

Epilepsy

12

1

1

 7.7610

 0.021

 0.012

11

13

2

Corpus callosum

8

0

0

 7.3913

 0.025

 0.022

15

14

3

Prominent CSF spaces

6

0

0

5.2174

0.074

 0.139

17

14

3

Other brain anomalies

6

2

1

0.9133

0.633

0.608

17

12

2

Micropenis

8

0

1

 5.1198

0.077

0.057

10

8

2

Cryptorchidism

8

0

0

6.7513

0.034

0.041

10

8

3

Cataract

1

1

0

0.3138

0.855

 1.000

22

13

3

Strabismus

10

2

0

 4.9216

 0.085

 0.103

13

12

3

Nystagmus

4

1

0

 1.2990

 0.522

 0.755

19

13

3

hearing loss

4

0

0

3.2850

0.193

 0.361

19

14

3

total=

23

14

3

In conclusion, even if reviewer’s comments were very useful, performing calculations we realized that  we do not think it is sensible adding them in the main text, not only from a statistical point of view and but also the difficulties deriving the transmission of the message that the number are providing.

Round 2

Reviewer 1 Report

I think the choice of the authors to present the paper as a review is more appropriate.

Reviewer 2 Report

I have already gone through the revision of the paper. As far as I'm concerned, the authors have addressed my comments and concerns appropriately.